# Association between Sarcopenia, Falls, and Cognitive Impairment in Older People: A Systematic Review with Meta-Analysis

**DOI:** 10.3390/ijerph20054156

**Published:** 2023-02-25

**Authors:** Jack Roberto Silva Fhon, Alice Regina Felipe Silva, Eveline Fontes Costa Lima, Alexandre Pereira dos Santos Neto, Ángela Maria Henao-Castaño, Elizabeth Fajardo-Ramos, Vilanice Alves Araújo Püschel

**Affiliations:** 1Graduate Program in Adult Health Nursing, Nursing School, University of São Paulo, São Paulo 05403, Brazil; 2Brazilian Center for Evidence-Based Healthcare, A JBI Centre of Excellence, São Paulo 05403, Brazil; 3Facultad de Enfermería, Universidad Nacional de Colombia, Bogotá 11001, Colombia; 4Facultad de Ciencias de la Salud, Universidad del Tolima, Ibagué 730006299, Colombia

**Keywords:** cognitive dysfunction, falls, sarcopenia, elderly, review

## Abstract

With the aging process, elderly people present changes in their bodies that can lead them to suffer several geriatric syndromes. The present study aimed to analyze and synthesize the literature produced concerning the association of sarcopenia with falls in elderly people with cognitive impairment. This is a systematic review study on etiology and risk, conducted according to the JBI methodology using the Medline (Pubmed), Cinahl, Embase, Scopus, and Web of Science databases. The gray literature search was conducted in the CAPES Brazilian Digital Library of Theses and Dissertations, Google Scholar, Networked Digital Library of Theses and Dissertations (NDLTD), EBSCO Open Dissertations, DART-e, and ACS Guide to Scholarly Communication. The identification of the association between the variables was extracted from the articles themselves (Odds Ratio and the 95% Confidence Intervals). Four articles published between 2012 and 2021 were included in this review. A prevalence of falls was identified, ranging from 14.2% to 23.1%, of cognitive impairment ranging from 24.1% to 60.8%, and of sarcopenia ranging from 6.1 to 26.6%. The meta-analysis found that elderly people with cognitive impairment who suffer falls are at a 1.88 times greater risk of presenting sarcopenia (*p* = 0.01). There is evidence of an association between the variables, but it is necessary to conduct follow-up studies to support this association as well as other factors that may influence the senescence and senility process.

## 1. Introduction

A report published by the World Population Prospects in 2019 estimates that by 2050 there will be twice as many people over the age of 65 than children under the age of 5 [1]. The number of people aged 60 and over will rise from 11% to 22%, i.e., from approximately 605 million to 2 billion over half a century [2].

The aging process is dynamic, gradual, natural, and inevitable [3]; it encompasses biological, physical, and psychological changes in individuals in continuous interaction with their social, economic, and cultural lives [4]. This process is driven by interconnected mechanisms that lead to the emergence of hallmark phenotypes, including changes in body composition, imbalances in energy production and use, homeostasis dysregulation, neurodegeneration, and loss of neuroplasticity [5].

As people age, their cell and tissue functions in all body systems are reduced, and changes in health ensue [6]. In this context, the chronic pro-inflammatory state is low-grade as one ages, triggering a risk factor for multiple morbidities, physical and cognitive impairment, sarcopenia, frailty, and death [5].

Elderly people can present progressive and generalized loss of muscle mass with a corresponding decrease in muscular and physical performance, known as sarcopenia. The European consensus defines it as a syndrome marked by a gradual and generalized loss of skeletal muscle mass and strength, with a risk of adverse outcomes such as poor quality of life, falls, fractures, and increased mortality [7].

Sarcopenia was originally defined as a loss of muscle mass associated with aging. In the last decade, however, it has become a broader term, often used to define the age-related loss of muscle mass and strength. Longitudinal studies concerning muscle mass and strength have shown that in older people muscle strength declines significantly faster than muscle mass, suggesting that muscle mass suffers with aging and that gaining muscle mass in itself does not prevent the decline in muscle strength. Furthermore, it is clear that sarcopenia alone is not a reliable predictor of functional decline and death, in contrast to decreased muscle strength, which has been associated with such outcomes [8].

The prevalence of sarcopenia is estimated to range from 5% to 13% in people aged 60 to 70 years and from 11% to 50% in people aged over 80 years [9]. The World Health Organization (WHO) estimates that more than 50 million people have sarcopenia and that this number will rise to more than 200 million in the next 40 years [10].

Considering the aging of the world population and the estimates of the significant increase in the elderly population, the occurrence of falls in the elderly population is expected to increase, which is considered a public health problem. The WHO defines a fall as an event that results in a person inadvertently hitting the ground, floor, or another lower level [11]. A study conducted with 356 elderly people from a community has shown that the prevalence of falls associated with sarcopenia can reach 90.7% [12] and that it can be caused by a decreased cognitive status [13].

Cognition maintenance in the elderly hinges on physiological and environmental factors, which may be related to age, living alone, a low level of education [9], diseases, social support, mood, the presence of other geriatric syndromes, economic status, and the location of residence [14].

A systematic review that related sarcopenia with cognitive impairment showed that the combined odds ratios for cognitive impairment in patients with sarcopenia compared to patients without sarcopenia were 2.85 (95% CI: 2.19–3.72) in the unadjusted analysis and 2.25 (95% CI: 1.70–2.97) in the adjusted meta-analysis [15], which related sarcopenia to falls, and the authors identified a significantly higher risk of falls and fractures in sarcopenic elderly people compared to non-sarcopenic individuals [16].

A preliminary search was carried out in databases such as the PROSPERO, MEDLINE, Cochrane Search Database of Systematic Reviews, and JBI Database of Systematic Reviews e Implementation Reports with the objective of identifying existing protocols or ongoing studies; however, no studies were found correlating sarcopenia, falls, and cognitive impairment. In this sense, the present review aimed to analyze and synthesize the literature produced concerning the association of sarcopenia with falls in elderly people with cognitive impairment.

## 2. Materials and Methods

This is a systematic review study on etiology and risk, conducted according to the JBI methodology, carried out in eight steps, namely: (1) review title; (2) objective and guiding question; (3) introduction (background); (4) inclusion criteria; (5) methods (search strategy, critical assessment, study selection, and data synthesis); (6) results; (7) discussion; and (8) conclusion and recommendations [17]. The guidelines of the Preferred Reporting Items for Systematic reviews and Meta-Analyses—PRISMA [18] were followed for write the article. This systematic review was registered in PROSPERO under CDR42022335131.

To develop the review question, the PEO mnemonic was used, where P = population (elderly people with cognitive impairment), E = Exposure of interest (sarcopenia), and O = Outcome (fall), i.e., “What is the association of sarcopenia with falls in elderly people with cognitive impairment in different contexts?”.

The inclusion criteria were the following: studies with elderly people aged with cognitive impairment ≥60 years, regardless of gender, ethnicity, social status, presence of comorbidities, location of residence, and in different contexts (hospital, home, and Long-Stay Institution for the Elderly). The age of 60 years or older was considered based on the WHO [19].

The definition of sarcopenia used in this review stems from the European working Group on Sarcopenia in Older People 2 (EWGSOP2), which considers it a muscle disease and muscular insufficiency, in which low muscle strength is the main determinant for diagnostic investigation, superseding low muscle mass [7]. In contrast, The Asian Working Group for Sarcopenia (AWGS) defines it as an age-related loss of muscle mass in addition to low muscle strength and/or poor physical performance [20]. Furthermore, a fall is defined as an event that results in a person inadvertently hitting the ground or another level [11].

Observational, prospective, and retrospective (longitudinal) cohort studies; case–control studies; cross-sectional analytical studies; experimental studies, such as randomized clinical trials, were included. Studies that established an association between sarcopenia and falls as measured by the number of such events in the past 6 or 12 months were also included. There were no publication dates or language restrictions. Systematic or integrative literature review studies were excluded.

The search strategy aimed to locate published and unpublished studies (gray literature) and was performed in three stages. In the first stage, an initial search was conducted in the Medline and CINAHL databases (Appendix A), analyzing the text words in the titles and abstracts in a sample of studies considered relevant to the review question.

In the second step, the search was carried out using all the words of the indexed controlled and uncontrolled descriptors in all the databases. In the third step, a search for additional studies was performed by reviewing the references of all articles identified in the second search and in the gray literature. The selection of the studies was performed by two independent reviewers. If disagreements could not be settled between the two reviewers, a third reviewer was involved to settle the disagreements.

The search in the databases was performed on 17 July 2022, and Table 1 presents the search strategy developed for use in the varying databases, such as Medline (Pubmed), Cinahl, Embase, Scopus, and Web of Science. The gray literature search was performed in the CAPES Brazilian Digital Library of Theses and Dissertations, Google Scholar, Networked Digital Library of Theses and Dissertations (NDLTD), EBSCO Open Dissertations, DART-e, and ACS Guide to Scholarly Communication. Furthermore, in Table 1, the number of files retrieved in the search process for each database searched appears.

After identifying the studies in the databases, the Rayyan [21] technological platform was used to carry out the selection of studies. Two reviewers blindly selected the studies applying the inclusion and exclusion criteria by reading the titles and abstracts, and in the case of disagreement, a third reviewer evaluated the article under discussion to identify whether it met the inclusion criteria together with the two reviewers from the previous step using the PRISMA [18]. The articles were read in full and exhaustively to check if they addressed the review question.

For data extraction from the studies, the JBI SUMARI [22] instrument was used, and the following data were extracted: author, year, journal, method, study design, study site, participants, recruitment procedure used, study duration, exposure of interest (independent variable—sarcopenia), outcome (dependent variable—fall), outcome measurements, data analysis method, and study results, appropriate measurements for effect size such as hazard ratio, relative risk ratio, odds ratio, *p*-value, and 95% confidence intervals.

The information was presented in the form of a narrative synthesis, employing quantitative data descriptive analysis and statistical analysis via a meta-analysis of the Odds Ratio (OR) association measurement. For data analysis, the R software (version 3.4.3) was used, in addition to the Metafor 2.0 meta-analysis package. The forest-type plots present the observed measurements observed (OR) on the *x*-axis, and the Confidence Interval (CI) is within the estimated limits, being equal to ±1.96 of the Standard Error.

The evaluation of the methodological quality was carried out via the JBI Critical Appraisal for Cohort Studies, Case–Control Studies, Case Reports, and Cross-Sectional Analytical Studies. The selected studies were evaluated by two independent reviewers, using the JBI tools of extraction and critical appraisal of the methodological quality of the studies.

The reviewers discussed the results of the individual critical appraisals, as well as the positive and negative aspects of each study, in order to compose the final appraisal scenario.

## 3. Results

In the 13 databases used, 439 files were identified. After reading the titles and abstracts and removing duplicates and studies that failed to meet the inclusion criteria, 27 studies remained. Of these, seven did not have their full text available, and 20 were retrieved for reading in full. Seven were removed for excluding elderly people with cognitive impairment, five for not investigating the occurrence of falls, two for not diagnosing sarcopenia, one for not assessing cognitive status, and one for being a secondary study. Thus, four studies were selected as eligible for inclusion in this systematic review (Figure 1).

The four studies included were published in 2021 [23,24], 2020 [25], and 2012 [26]. Regarding the language, three were published in English [24,25,26], and one was published in Portuguese [23].

The studies were carried out in Italy [26], China [25], South Korea [24], and Brazil [23], and they were published in Europe [26], the United States [25], Japan [24], and Brazil [23], namely: Clinical Nutrition [26], Medical Science Monitor [25], Geriatrics & Gerontology International [24], and in the Theses and Dissertations Catalog of CAPES [24]. Regarding the methodology used, two studies are cross-sectional [23,25], and two are longitudinal prospective cohort studies [24,26].

A total of 3283 elderly people participated in the studies, ranging from 260 [26] to 2028 [24]. The age groups of the elderly people included ranged from 60 years [23] to 70 and older [24] and to 80 years and older [25,26].

Regarding falls, it was found that the lowest prevalence was 14.2% [26], and the highest was 23.1% [24]. For the identification of cognitive impairment, the following instruments were used: Mini Mental State Examination (MMSE) [23,24], Montreal Cognitive Assessment (MoCA) [25], and Minimum Data Set for Home Care (MDS-HC) [26]. The prevalence of cognitive impairment ranged from 24.1% [24] to 60.8% [25]. For the diagnosis of sarcopenia, several different criteria have been adopted, namely: AWGS [24,25], EWGSOP [26], and EWGSOP2 [25]. The prevalence of sarcopenia ranged from 6.1% [24] to 26.6% [25] (Table 2).

When extracting the results regarding the association measurements, it was found that three studies had identified an association between sarcopenia and falls in elderly people with cognitive impairment. The association measurements used in the studies are different, and no association was found between the study variables (Table 3).

Regarding the values from the studies that used OR, which are cross-sectional studies, as an association measurement, the interpretation is that the elderly who present cognitive impairment and suffer falls are at 1.88 times increased risk of presenting sarcopenia compared to those who do not present cognitive impairment, which is statistically significant (*p* = 0.01). It was found that there is no heterogeneity between the values from the studies (*I*^2^ = 0%), which is significant with *p* = 0.013 (Figure 2).

The designs of the included studies were cross-sectional and longitudinal, with results relevant to the objective and question of this systematic review. Table 4 summarizes the methodological quality of the cross-sectional studies.

In the evaluation of the cross-sectional studies, most criteria were met. Three of the addressed criteria had a frequency of 50% in questions Q5 (Were confounding factors identified?), Q6 (Were strategies to address confounding factors stated?), and Q8 (Was a suitable statistical analysis used?) in the study by Santos 2021 [23], lowering the general evaluation of ConQual.

Table 5 shows the evaluation of the methodological quality of the longitudinal studies. There were two criteria with 50% scores for questions Q7 (Were outcomes measured in a valid and reliable way?) and Q8 (Was the follow-up time reported and sufficient to be long enough for outcomes to occur?). Q10 (Were strategies used to address incomplete follow-up?) had a 0% score.

## 4. Discussion

In this systematic review, four studies that answered the proposed study question were identified. The prevalence of falls, cognitive impairment, and sarcopenia and the interaction of these three events in the elderly population were identified.

Analyzing the individual results, regarding cognitive impairment, a prevalence of 24.1% [24] to 60,8% [25] was found using several screening instruments, such as the MMSE, MoCA, and MDS-HC. Aging is associated with the decline of various cognitive processes, such as episodic memory, attention, and executive function, which rely on the hippocampus and prefrontal cortex [27].

Regarding falls, the lowest prevalence was 14.2% [26], and the highest was 23.1% [24]. Falls are a public health problem, which can be caused by intrinsic, extrinsic, and behavioral factors, and can lead to changes in functionality, hospitalization, and death [28].

The prevalence of sarcopenia in the studies ranged from 6.1% [24] to 26.6% [25]. It is caused by the loss of skeletal muscle mass followed by reduced muscle strength or physical function [29], which impacts quality of life and also increases the risk of death [30]. In this sense, as the world population ages, the prevalence gradually increases [29]. However, the limited number of studies found that combine these three events indicates the importance of conducting research aimed at screening the elderly population by healthcare professionals.

In this review, it was found that there was a relationship between the variables researched. Studies have shown that sarcopenia and cognition share risk factors, such as presenting depressive symptoms [24], malnutrition [31], and global disability, which includes physical, motor, and gait components [32], which lead the elderly to suffer from falls; however, this association is still unclear.

Chronic inflammation related to the aging process, which is defined by elevated levels of interleukin-6 and tumor necrosis factor-α, has been a causal factor of decreased cognitive performance and the presence of sarcopenia in the elderly population [33]. Furthermore, excessive oxidative stress related to chronic diseases can lead to loss of skeletal muscle mass, leading the elderly population to be considered sarcopenic [34] and experience possible falls.

There is a need to have detailed information that directs adequate healthcare, which requires the implementation of available measures for detailed screening of the elderly, with the use of validated instruments such as the MMSE, the use of biomarkers such as hemoglobin, urea, creatinine, calcium, phosphorus, albumin, C-reactive protein (CRP), HbA1c, and fibrinogen, which direct the identification of sarcopenia and the identification of falls for a diagnostic, safe, and evidence-based practice [8].

It is worth noting that age-related decline or loss of muscle strength, mass, and function are risk factors for an increase in falls associated with a decrease in physical function. This is due to the fact that muscle function plays a more important role than muscle mass. Muscle function is consistently associated with falls [35].

Gadelha and colleagues [36] have found an association between the various stages of sarcopenia and fall events, showing that the severity of the condition, i.e., sarcopenia, is related to reduced balance and increased risk of falls as evaluated by a strength platform and the Quickscreen fall risk assessment, respectively, but cognitive impairment in the elderly was not evaluated.

Falls are one of the leading causes of morbidity and mortality in the elderly. Currently, the Stopping Elderly Accidents, Deaths and Injuries (STEADI) initiative of the Centers for Disease Control and Prevention (CDC) aims to reduce the risk of falls among the elderly. Healthcare professionals need to identify and assess this type of risk using simple, low-cost measures, such as: measuring orthostatic blood pressure (lying and standing position), questioning patients about potential hazards in their household, such as the use of rugs, slippery bathtub floors, and so on, identifying medications that increase the risk of falls, and assessing the elderly patients’ gait, strength, and balance [37].

Among the limitations of the studies, the differing approaches used to define cognitive impairment, falls, and sarcopenia are identified. Of the four studies that are part of the review, a meta-analysis was only performed in two studies using the cross-sectional method due to the use of different association measurements. Furthermore, there was a lack of association in prospective studies in this review. Although few studies have studied this association between the study variables, it is important for healthcare professionals to be aware of it and evaluate it constantly.

## 5. Conclusions

This review has found four studies that studied the study variables using different methodologies. According to the analysis of these studies, the authors used different association measurements, and it was possible to perform a meta-analysis with two cross-sectional studies that used OR where an association was found, but without the existence of heterogeneity among the studies.

It is necessary to conduct follow-up studies in order to identify the cause-effect relationship between the different geriatric syndromes in the elderly with the objective of providing health interventions for this population regarding the outcome of cognitive impairment, falls, and sarcopenia, with the use of instruments validated for the study population.

This study indicates the need for more scientific production, especially regarding the improvement of the complexity and application of detailed clinical exams with the participation of multidisciplinary teams based on the relevance of the problem, which is a predictor of unfavorable clinical outcomes and decreased quality of life in the studied population.

## Figures and Tables

**Figure 1 ijerph-20-04156-f001:**
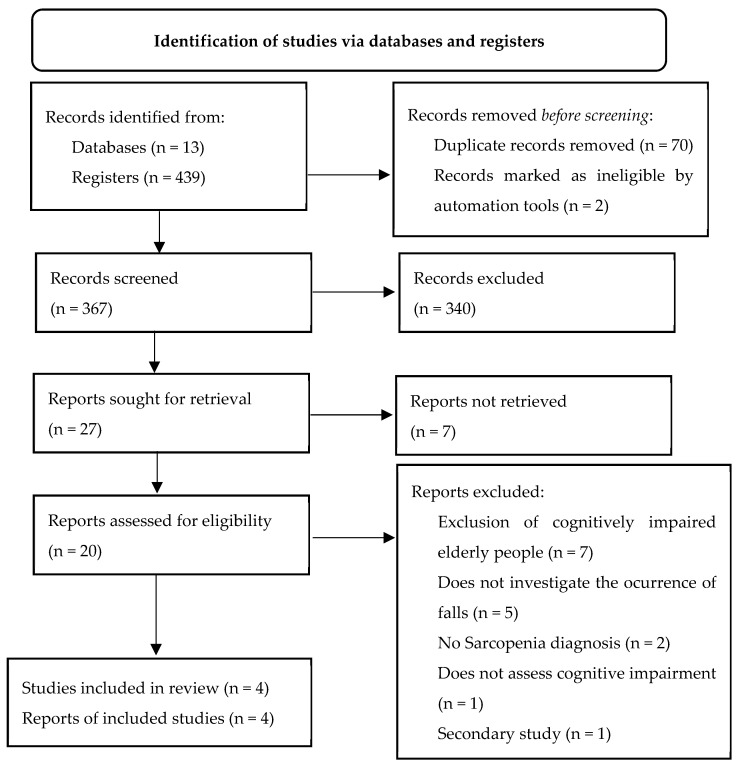
PRISMA Flowchart of study identification and selection. São Paulo, SP, Brazil, 2022.

**Figure 2 ijerph-20-04156-f002:**
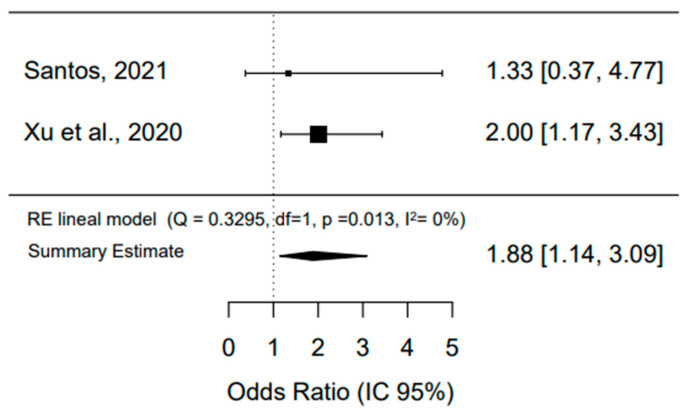
The association between cognitive impairment, falls, and sarcopenia in the elderly, São Paulo, SP, Brazil, 2022 [23,25].

**Table 1 ijerph-20-04156-t001:** Search strategy in the various databases, São Paulo, SP, Brazil, 2022.

Database	Search Strategy	Results
BVS	((mh:(“Acidentes por Quedas”)) OR (“Acidentes por Quedas” OR “Accidental Falls” OR “Accidentes por Caídas”)) AND ((mh:(Idoso)) OR (Idoso OR Aged OR Anciano)) AND ((mh:(Sarcopenia)) OR (Sarcopenia)) AND ((mh:(“Disfunção Cognitiva”)) OR (“Disfunção Cognitiva” OR “Cognitive Dysfunction” OR “Disfunción Cognitiva”))	9
PUBMED	(((“Accidental Falls”[Mesh] OR “Accidental Falls” OR “Accidental Fall”) AND (“Aged”[Mesh] OR Aged)) AND (“Sarcopenia”[Mesh] OR Sarcopenia)) AND (“Cognitive Dysfunction”[Mesh] OR “Cognitive Dysfunction”)	7
CINAHL	((MH “Accidental Falls”) OR “Accidental Falls” OR “Accidental Fall”) AND ((MH “Aged”) OR Aged) AND ((MH “Sarcopenia”) OR Sarcopenia) AND ((((MH “Cognition Disorders”) OR “Cognitive Disorders” OR “Cognitive Dysfunction”)) OR ((MH “Cognitive Aging”) OR “Cognitive Aging”)))	11
EMBASE	(falls OR “accidental falls” OR falling) AND (Aged) AND (Sarcopenia) AND (“cognitive defect” OR “cognitive dysfunction” OR “cognitive disorder” OR “cognitive disorders”)	82
SCOPUS	((fall*) AND (aged) AND (sarcopenia) AND ({cognitive defect} OR {cognitive dysfunction} OR {cognitive disorder} OR {cognitive disorders}))	72
Web of Science	(fall*) AND (Aged) AND (Sarcopenia) AND (“cognitive defect” OR “cognitive dysfunction” OR “cognitive disorder” OR “cognitive disorders”)	9
Google Scholar	(“Accidental Falls”) AND (Aged) AND (Sarcopenia) AND (“cognitive defect” OR “cognitive dysfunction” OR “cognitive disorder” OR “cognitive disorders”)	97
CAPES Theses and Dissertations	(Queda OR Quedas) AND Idoso AND Sarcopenia	29
Networked Digital Library of Theses and Dissertations (NDLTD)	(“Accidental Falls”) AND (Aged) AND (Sarcopenia)	38
EBSCO Open Dissertations	(“Accidental Falls”) AND (Aged) AND (Sarcopenia)	51
DART-e	(Fall OR Falls) AND (Aged) AND (Sarcopenia)	3
Brazilian Digital Library of Theses and Dissertations	(Fall OR Falls) AND (Aged) AND (Sarcopenia)	18
ACS Guide to Scholarly Communication	(Fall OR Falls) AND (Aged) AND (Sarcopenia)	13
Total Records		439

**Table 2 ijerph-20-04156-t002:** The estimated prevalence of cognitive impairment, falls, and sarcopenia from the studies included in the systematic review, São Paulo, SP, Brazil, 2022.

Author, Year	Population	Age, Sex	Type of Study	Instruments	Prevalence
Santos, 2021 [23]	413 elderly patients receiving care at the geriatrics and gerontology outpatient unit	≥60 years old, of both sexes, with a prevalence of females	Cross-sectional	EWGSOP2 criteria *** (handgrip strength—SAEHAN digital dynamometer, Gait Speed Test, Body Mass Index) Report of falls over the last 3 months Mini Mental State Examination (MMSE)	Sarcopenia - Probable: 57.9% - Confirmed: 6.1% - Severe: 6.8% Falls: 18.4% Cognitive Impairment: 25.7%
Lee et al., 2021 [24]	2028 elderly outpatients	≥70 years, of both sexes	Retrospective longitudinal	AWGS Criteria * 2014 and 2019 (Dual-energy X-ray absorptiometry, digital handgrip dynamometer, usual gait speed of 4 m) Report of falls over the last 12 months MMSE	Sarcopenia: 17.5% (AWGS 2019) 9.1% (AWGS 2014) Falls: 23.1% of elderly people with sarcopenia Cognitive Impairment: 24.1%
Xu et al., 2020 [25]	582 elderly people from the community	≥80 years, of both sexes, with a prevalence of females	Cross-sectional	AWGS Criteria * (Handgrip strength—dynamometer, usual gait speed, electrical bioimpedance) Report of falls over the last 12 months Montreal Cognitive Assessment (MoCA)	Sarcopenia: 26.6% Falls: 18.1% Cognitive Impairment: 60.8%
Landi et al., 2012 [26]	260 elderly people from the community	≥80 years, of both sexes, with a prevalence of female	Prospective longitudinal	EWGSOP Criteria ** (middle arm muscle circumference, usual gait speed, handgrip strength—dynamometer) Report of falls within 24 months of follow-up Minimum Data Set for Home Care (MDS-HC)	Sarcopenia: 25.4% Falls: 14.2% Cognitive Impairment: 14.2%

* AWGS: Asian Working Group for Sarcopenia. ** EWGSOP: European Working Group on Sarcopenia in Older People. *** EWGSOP2: European Working Group on Sarcopenia in Older People 2.

**Table 3 ijerph-20-04156-t003:** Association measurements used and analyzed in the studies included in the systematic review, São Paulo, SP, Brazil, 2022.

Author	Association Measurement	*p*-Value
Santos, 2021 [23]	OR 1.333 (0.372–4.775)	0.658
Lee et al., 2021 [24]	-	<0.001
Xu et al., 2020 [25]	OR 2.00 (1.17–3.43)	-
Landi et al., 2012 [26]	HR 3.23(1.25–8.29)	-

OR = Odds Ratio; HR = Hazard Ratio.

**Table 4 ijerph-20-04156-t004:** Critical evaluation of the cross-sectional studies included, São Paulo, SP, Brazil, 2022.

Studies	Q1	Q2	Q3	Q4	Q5	Q6	Q7	Q8
Santos, 2021 [24]	Y	Y	Y	Y	N	N	Y	N
Xu et al., 2020 [26]	Y	Y	Y	Y	Y	Y	Y	Y
Total %	100%	100%	100%	100%	50%	50%	100%	50%

Y = Yes; N = Not; Q1—Were the inclusion criteria in the sample clearly defined? Q2—Were the study subjects and setting described in detail? Q3—Was the exposure measured in a valid and reliable way? Q4—Were objective and standard criteria used for the measurement of the condition? Q5—Were confounding factors identified? Q6—Were strategies to address confounding factors stated? Q7—Were the results measured in a valid and reliable way? Q8—Was a suitable statistical analysis used?

**Table 5 ijerph-20-04156-t005:** Critical evaluation of the longitudinal studies included, São Paulo, SP, Brazil, 2022.

Studies	Q1	Q2	Q3	Q4	Q5	Q6	Q7	Q8	Q9	Q10	Q11
Lee et al., 2021 [25]	Y	Y	Y	Y	Y	Y	Y	P	Y	NP	Y
Landi et al., 2012 [27]	Y	Y	Y	Y	Y	Y	P	Y	Y	NP	Y
Total %	100%	100%	100%	100%	100%	100%	50%	50%	100%	0%	100%

Y = Yes; P = Partially; NP = None of the previous; Q1—Were both groups similar and recruited from the same population? Q2—Were exposures measured similarly to assign people to the exposed and unexposed groups? Q3—Was the exposure measured in a valid and reliable way? Q4—Were confounding factors identified? Q5—Were strategies for addressing confounding factors stated? Q6—Were the groups/participants free of the outcome at the beginning of the study (or at the time of exposure)? Q7—Were outcomes measured in a valid and reliable way? Q8—Was the follow-up time reported and sufficient to be long enough for outcomes to occur? Q9—Was follow-up complete? If not, were the reasons for the loss of follow-up described and explored? Q10—Were strategies used to address incomplete follow-up? Q11—Was a suitable statistical analysis used?

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
