# Peer review of "Association between Sarcopenia, Falls, and Cognitive Impairment in Older People: A Systematic Review with Meta-Analysis"

_ijerph, 2023, doi:10.3390/ijerph20054156_

Round 1
Reviewer 1 Report
Thank you for the opportunity to review this systematic review investigating the association of sarcopenia with falls in elderly people with cognitive impairment. The meta-analysis found that elderly people with cognitive impairment who suffer falls are at a 1.88 times greater risk of presenting sarcopenia (p = 0.01). This reviewer has certain questions, which should be addressed by authors prior to publication.
You have described that “the present review aimed to analyze and synthesize the literature produced concerning the association of sarcopenia with falls in elderly people with cognitive impairment.”. So, this review should analyze the association between sarcopenia and falls among elderly people with cognitive impairment. However, Xu’s study included in the meta-analysis (Figure 2) did not show the odds ratio that the association between sarcopenia and falls among people with cognitive impairment were analyzed. They said that sarcopenia was significantly and independently associated with cognitive decline or falls. Therefore, it is not appropriate that Xu’s study is included in this meta-analysis. There are same concerns in other studies (26,27). They also did not compare elderly people with cognitive impairment to those without cognitive impairment. It is not clear what associations were observed in the previous studies in the Table 1. More explanation is needed.
It seems that this study has followed PRISMA guideline, was this study registered PROSPERO?
On p.2, line 72 to 75, “level of education” is repeated in this sentence.
In the figure 1, some sentences were not included in squares in the second low.
Several database were used to search articles. It would be better to report how many articles were identified in each database.
In the Chart 2, the Instruments low in the Santos’s study, Muscle Mass, but not “Body Mass Index” would be correct.
Author Response
Revisor 1
Thank you for the opportunity to review this systematic review investigating the association of sarcopenia with falls in elderly people with cognitive impairment. The meta-analysis found that elderly people with cognitive impairment who suffer falls are at a 1.88 times greater risk of presenting sarcopenia (p = 0.01). This reviewer has certain questions, which should be addressed by authors prior to publication.
You have described that “the present review aimed to analyze and synthesize the literature produced concerning the association of sarcopenia with falls in elderly people with cognitive impairment.”. So, this review should analyze the association between sarcopenia and falls among elderly people with cognitive impairment.
However, Xu’s study included in the meta-analysis (Figure 2) did not show the odds ratio that the association between sarcopenia and falls among people with cognitive impairment were analyzed.
They said that sarcopenia was significantly and independently associated with cognitive decline or falls. Therefore, it is not appropriate that Xu’s study is included in this meta-analysis. There are same concerns in other studies (26,27). They also did not compare elderly people with cognitive impairment to those without cognitive impairment. It is not clear what associations were observed in the previous studies in the Table 1. More explanation is needed.
Answer: The reviewer is misreading the figure, as it is not a meta-analysis of effectiveness studies, but of risk etiology. In a review, it is necessary to include not only studies that showed an association, but also studies that did not show these results.
The studies that are going into this review, the aim in the studies was not to compare both groups. The studies compared in the general study population.
It seems that this study has followed PRISMA guideline, was this study registered PROSPERO?
Answer: PRISMA is not a methodological framework for a systematic review, it is a guide for writing a review publication. The methodological framework used in this review is the JBI being a systematic review of risk etiology. And the JBI methodology recommends registering the revision in different databases, one of them being PROSPERO, in which the revision was registered.
On p.2, line 72 to 75, “level of education” is repeated in this sentence.
Answer: Correction has been performed.
In the figure 1, some sentences were not included in squares in the second low.
Answer: According to PRISMA indications, the flowchart must be published as it appears, even if the indicated rectangles are not used. To avoid confusion for the reader, this information has been left blank.
Several database were used to search articles. It would be better to report how many articles were identified in each database.
Answer: This information appears in Table 1
In the Chart 2, the Instruments low in the Santos’s study, Muscle Mass, but not “Body Mass Index” would be correct.
Answer: Among the instruments used by the author and the Body Mass Index, in this sense the information is correct.
Reviewer 2 Report
This is a well written, informative article. My comments are of readability as of the tables were difficult to read. Chart 1 was not clearly stated, I recommend clarifying Chart 1, compressing some definitions. Figure 1 was also confusing in that the identification of studies via other methods all had n of "0", could this be stated in a clearer manner? The other tables and charts were clearer.
Author Response
Revisor 2
This is a well written, informative article. My comments are of readability as of the tables were difficult to read. Chart 1 was not clearly stated, I recommend clarifying Chart 1, compressing some definitions.
Answer: Explanation of chart 1 was carried out
Figure 1 was also confusing in that the identification of studies via other methods all had n of "0", could this be stated in a clearer manner? The other tables and charts were clearer.
Answer: According to PRISMA indications, the flowchart must be published as it appears, even if the indicated rectangles are not used. To avoid confusion for the reader, this information has been left blank.

Reviewer 3 Report
This is an interesting systematic review and metanalysis about the relationship between sarcopenia, falls and aging in the elderly population, as is expected there an increased risk of falls associated, interesting using very few clinical studies and finally used only 2 cross-section studies to do metanalysis, only minor comments:
-Add as a limitation the lack of association in prospective studies
-Correct minor English sentences as in line 151 dates analysis.
-It is better to include abbreviations of the neuropsychological tests in the text and not only in Tables.
-In the Discussion, the author didn't mention any line about preventing falls in this clinical situation and very little about the comorbidities mentioned in the four papers analyzed.
Author Response
Revisor 3
This is an interesting systematic review and metanalysis about the relationship between sarcopenia, falls and aging in the elderly population, as is expected there an increased risk of falls associated, interesting using very few clinical studies and finally used only 2 cross-section studies to do metanalysis, only minor comments:
-Add as a limitation the lack of association in prospective studies
Answer: Correction has been performed.
-Correct minor English sentences as in line 151 dates analysis.
Answer: Correction has been performed.
-It is better to include abbreviations of the neuropsychological tests in the text and not only in Tables.
Answer: Correction has been performed.
-In the Discussion, the author didn't mention any line about preventing falls in this clinical situation and very little about the comorbidities mentioned in the four papers analyzed.
Answer: Situation and very little about the comorbidities mentioned in the four papers analyzed.

Round 2
Reviewer 1 Report
This reviewer could not find your PROSPERO, no. CDR42022335131.
Author Response
The review protocol is registered but how it appears in the screenshots of prospero has not yet been published.
I send the pdf of the publication of the protocol
Yours sincerely
